# Hazard Analysis and Risk-Based Preventive Controls (HARPC): Current Food Safety and Quality Standards for Complementary Foods

**DOI:** 10.3390/foods10092199

**Published:** 2021-09-16

**Authors:** Sargun Malik, Kiruba Krishnaswamy, Azlin Mustapha

**Affiliations:** 1Division of Food, Nutrition and Exercise Sciences, University of Missouri, Columbia, MO 65211, USA; smvkf@umsystem.edu (S.M.); MustaphaA@missouri.edu (A.M.); 2Department of Biomedical, Biological and Chemical Engineering, University of Missouri, Columbia, MO 65211, USA

**Keywords:** complementary food, infants, FSMA, HACCP, regulations

## Abstract

Food safety is imperative, especially for infants and young children because of their underdeveloped immune systems. This requires adequate nutritious food with appropriate amounts of macro- and micronutrients. Currently, a well-established system for infant food is enforced by the regulatory bodies, but no clear system exists for complementary food, which is consumed by children from the age of 6 month to 24 months. As the child grows beyond 6 months, the need for nutrients increases, and if the nutritional needs are not fulfilled, it can lead to health problems, such as stunted growth, weak immune system, and cardiovascular diseases. Hence, it is important to have regulatory bodies monitoring complementary food in a similar capacity as is required for infant formula. The objective of this review is to provide an overview of the existing regulatory bodies, such as the Codex Alimentarius, International Standard Organization (ISO), Food and Drug Administration (FDA), etc., and their regulations specifically for infant formula that can be adopted for complementary foods. This study focuses on the development of a hazard analysis and risk-based preventive controls (HARPC)-based food safety plan to ensure safe food processing and prevent any possible outbreaks.

## 1. Introduction

Nutrition at an early stage of life lays a foundation for the optimal growth and development of the human body. Breast milk is the primary food source for children under the age of six months, but as the child grows older, nutrient needs increase. Hence, a complementary food is required to fulfill all nutritional requirements, including energy, proteins, carbohydrates, fats, minerals, and vitamins [1]. If proper nutrients are not fed according to the child’s age, the child might suffer from problems, such as stunted growth, a weak immune system, cardiovascular diseases, etc. The foods required according to different age groups is illustrated in Table 1. Complementary food is usually based on staple diets having cereals and legumes as their main ingredient. For a good quality complementary food, there should be sufficiently high protein content, high caloric value per volume of food and adequate amounts of vitamins and minerals [2]. The other commonly used complementary foods include mashed fruits, vegetables, cooked grains, beans, etc. Currently, there are several commercially prepared complementary food brands available in the market that consist of these ingredients.

In the recent years, issues concerning contamination, fraud and adulteration of infant formula and complementary food have been reported to have increased [4]. Considering the higher risk of food-borne illness in children, the safety of food during its production, processing and handling practices plays a vital role in a child’s health [5]. In many countries, there are no specific regulatory requirements for complementary food. It is considered similar to any other food product, because it is regarded as not the sole part of the child’s diet [6,7]

Contamination can occur at numerous points of the food processing stage, i.e., before, during, and after processing. For example, at the point of raw material collection, transportation, cleaning, packaging, etc. Rigorous quality assurance (QA) and quality control (QC) processes can mitigate food safety incidents by having a good understanding of potential sources and routes of contamination [8]. Quality assurance includes a set of activities to ensure quality in whole production, distribution and procuring raw materials from suppliers safely. Quality control emphasizes the end product. Both processes oversee specific nutrient levels, essential testing for chemical residues, testing of microbiological pathogens and toxins, allergens, intentional and unintentional additives, environmental contaminants, labeling requirements and entire facility operational requirements [9].

Regulatory bodies are responsible for building laws and regulations to prevent the occurrence of food safety issues [10]. Countries that do not have the requisite infrastructure to build these laws and regulations follow the international food standards structured by the Food and Agriculture Organization (FAO) and World Health Organization (WHO) under the program known as Codex Alimentarius [11]. In the United States of America (USA), the food processing sector is controlled by state and federal agencies, wherein Hazard Analysis and Critical Control Point (HACCP) is the management system that helps in controlling microbiological, chemical, and physical hazards as well as improving the quality of products such as meat, poultry, seafood, and juice. Recently, the U.S governing bodies passed the Food and Drug Administration (FDA) Food Safety Modernization Act (FSMA). FSMA aims to help the regulatory bodies prevent the food safety issues instead of just focusing on the aftermath steps.

The objective of this paper is to provide an overview of the current food safety and quality standards required for complementary food. Regulations that exist for infant formula that could be adopted for complementary food. This study focuses on the need for more strict standards and aims to build a food safety plan based on HACCP and FSMA regulations for a nutritious ancient seed/grain-based complementary food product.

## 2. Microorganisms and Foodborne Illnesses Related to Infant Formula and Complementary Food

In the first few months after birth, the child is protected by maternal antibodies, but as the child grows older, the ability of the child to cope with pathogenic microorganisms is still in an emerging stage [12]. Food poisoning affects children more, as it can result in severe dehydration via vomiting and diarrhea. Hence, in the preparation of any infant formulations or complementary food, it is important to review the safety and suitability of the food product, including the processing techniques and ingredient quality [13]

Commercially prepared infant formulas and complementary foods are mostly brought in powdered form because it is convenient to use, has a longer shelf life and is easy to store. Hence, these powders are considered to be in the low moisture food category and are more susceptible to the growth of various pathogenic bacteria that can survive in dry and low-moisture environments [14]. Foodborne illness is usually associated with heat-labile micronutrients that are added after the heating step to keep the nutritional content compatible with standards set by regulatory agencies [15]. The major reasons for an outbreak are classified into three categories: biological, chemical, and physical hazards [16].

Infections caused by pathogenic *Cronobacter* spp. and *Cronobacter sakazakii* are of great public health concern. *C. sakazaki* is a member of the Enterobacteriaceae family and is a Gram negative, facultatively anaerobic, non-sporulating, motile rod-shaped bacterium. It is responsible for causing outbreaks linked with reconstituted powdered infant formula. It is particularly fatal for infants and creates health complications such as urinary tract infection, neonatal meningitis, sepsis, and seizures [17,18].

Microorganisms such as bacteria, viruses, and yeast are the major cause of biological hazards. Another major concern for infant food is the growth of *Salmonella*, which can lead to death of infants and is the reason behind several outbreaks. This bacterium can cause diarrhea, which can further progress into meningitis or bacteremia. The source of transmission can range from poor hygiene, contamination of raw materials, processing failures or even environmental contamination. There are several recorded outbreaks that are related to *Salmonella enterica* in countries like Spain, China, France, and Greece. One of the major outbreaks occurred in 2018, wherein 12 million cans of baby milk manufactured by a French company were recalled from over 83 countries [19,20].

*Clostridium botulinum* is a reason of concern for infant food, as this microorganism can cause a neuromuscular disorder called botulism. *C. botulinum* is capable of synthesizing seven neurotoxins, namely A to G, and can cause an imbalance in the intestinal microbiota leading to inflammatory bowel disease. The toxinfection is facilitated by the high pH and a microbiota development process that enables the spores to germinate and produce neurotoxins [21].

*Klebsiella pneumonia* can be grown in powdered infant formula and is classified as hazard category B. It has the potential to cause nosocomial infections like pneumonia, septicemia and urinary tract infections in individuals who have weak immune systems. The bacterium is strong enough to survive in the stomach environment and protect itself from natural defense cells; it can also resist antibiotics [22].

Another microorganism harmful for infants that can contaminate infant formula is *Staphylococcus aureus*. Some of the species of Staphylococcus are capable of producing exotoxins which can suppress the immune response and cause symptoms such as diarrhea, abdominal colic and vomiting. The source of contamination can be failures during processing or reconstitution of the product [23]. B. cereus is often the cause of contamination of powdered dairy products and can cause diarrheal and emetic syndromes. The food matrix of the formulas enables the growth of bacteria and production of spores, and, hence, even low-level contamination is a major concern for the health of children [24].

Presence of any chemical residue, such as pesticides, antibiotics, herbicides, insecticides, sanitizers etc., in food can lead to chemical hazards. These chemical residues should either be in a permissible amount that is set by the regulatory authorities or should not be found in the food product entirely. The sources of chemical hazards include environmental contamination, processing steps, additives, etc. In the case of baby food, heavy metals such as cadmium, lead, arsenic, and mercury are the major cause of concern. Exposure to these heavy metals can lead to severe health issues, such as brain damage, and lung and skin cancer [25]

In 2008, 300,000 people reported illness; 860 children were hospitalized, out of which 154 were classified as severe cases due to consumption of substandard milk and infant formula leading to kidney stone formation and kidney failures. The reason for the outbreak was melamine, which was added to formula make it appear that the product was higher in protein content because of its high nitrogen content. Melamine is not approved by food standard commissions. The outbreak prompted authorities to improve the safety standards, such that even low concentration of chemicals should be considered harmful [26].

Physical hazards include the discovery of foreign objects such as broken glass, plastic, stones, wood, or metal in the food product. The sources for physical hazards primarily include glass storage containers, packaging material, equipment, etc. Children are more prone to choking hazards. Physical hazards can lead to cuts in the throat and can also affect intestines [27]. The recent recalls and outbreaks based on these hazards have been described in Table 2.

## 3. Global Food Safety Standards for Complementary Foods: CODEX Alimentarius & ISO

The Codex Alimentarius defines the standards for follow-up formula as food made from either milk of cows or other animals, or from constituents of plant origin. Preparation should be carried out in a way that prevents any possible spoilage and contamination. The regulation describes that the product should be nutritionally sufficient to promote normal growth of the child with the help of essential nutrient contents of protein, fat, carbohydrates, vitamins, and minerals. All ingredients should be of good quality and appropriate for ingestion. The energy content of 100 mL of product should not be less than 60 kcal and should not be greater than 85 kcal. The essential nutrients required for complementary food are listed in Table 3. Other than the essential nutrients, these standards also enumerate the amount of food additives, including thickening agents, emulsifiers, pH adjustors, antioxidants, and flavors, as well as the permissible limits of pesticide residue and other contaminants. The standards also specify that after production, appropriate testing of the finished food product should be done in terms of pathogenic microorganisms which may represent hazards to health [11]. The preparation and packaging should be executed in accordance with the Code of Hygienic Practice for Infants and Young Children (CAC/RCP 66-2008).

The International Organization for Standardization (ISO) is a worldwide organization of national standard bodies. The aim of this organization is to develop standards in cooperation with international industry bodies which are relevant to the market and provide solutions to challenges at the global level. The ISO has strict standards for infant formula, considering its usage for a potentially vulnerable population. The different sections of the regulation mention the range of the essential macro- and micronutrients that should be present in the formula, as well as the appropriate method of quantification, as overviewed in Table 3 [31]. These quantification methods can also be adapted for complementary food in compliance with the content ranges mentioned in the Codex Alimentarius. Recently, private standards have also emerged and evolved [32]. The private global food safety standards currently available for the industry are summarized in Table 4. These private standards are now becoming increasingly important.

## 4. Food Safety Standards for Complementary Food in the United States

The FDA has no specific requirement for complementary food and includes it in general food products. However, it has robust infrastructure and regulations for infant formula, pertaining to its consumption during the critical period of growth of a child. These regulations, if applied to complementary food, can help prevent all types of hazards which have severe consequences for the health of children.

Within FDA, the Center for Food Safety and Applied Nutrition (CFSAN) is accountable for the regulation of infant formula along with the Office of Nutritional Product, Labelling and Dietary Supplements (ONPLDS) and Office of Food Additive Safety (OFAS). The Federal Food Drug Cosmetic Act (FFDCA) is a law that falls under United States Code and gives control to the FDA to manage food, drugs, cosmetics, and medical devices. FFDCA describes infant formula as a food consumed by children less than 12 months old that purports special dietary use for simulation of human milk. Under Section 412 of the FFDCA, prior to the marketing of a new formula, the formula should be registered. Subsequently, the FDA has to be notified if there are any changes in the formula. This section of the FFDCA also describes the responsibility of the FDA to monitor the implementation of all provisions. Every person responsible for either manufacture or distribution of the formula is required to adhere to the requirements stated in the regulation [33]. The code of federal regulations applicable to the handling of infant formula are shown in Table 5 and Figure 1 describes the by law arrangement of these regulations. These existing codes can be used to regulate complementary foods.

### Hazard Analysis and Critical Control Point (HACCP) and Food Safety Modernization Act (FSMA)

HACCP and FSMA are the food safety systems that enable the development of efficient controls for the production of food products so that there is a minimal occurrence food-related hazard [39]. These food safety systems also facilitate traceability and help in auditing the manufacturing process.

HACCP is a management system that addresses food safety from various aspects. It involves addressing food safety issues by analysis of biological, chemical and physical hazards in all sections of food manufacturing, including growing, harvesting, processing, and distributing [40]. The HACCP plan does not stand alone in a processing facility; it is built on other food safety plans, such as the prerequisite programs, which are designed to provide basic operating conditions for safe production of food, Good Manufacturing Practices (GMP), and sanitation standard operating procedures. The Current Good Manufacturing Practices (cGMPs) cover the in-process control systems, including controls to prevent contamination by workers, facilities, automatic equipment, utensils, ingredients, and containers. cGMPs also oversee adulteration from microorganisms, packaging material, labelling and the release of finished product. The minimum cGMPs that should be used in facilities for manufacturing, processing, packaging and holding of formula to ensure adequate retention of nutrients is also listed in the plan.

The preliminary tasks in development of the HACCP plan involves assembling the team, contemplation about the kind of food and its ingredients, and planning for processing and distribution with respect to its targeted consumer and intended use. Assembling the team involves collecting individuals who have the knowledge and experience with the target food product. These individuals have the responsibility to conduct the hazard analysis, identify potential hazards, suggest the controls and critical limits, frame procedures for monitoring and verification, recommend corrective actions and validate the food safety plan. The identified hazards are further categorized into their likeliness of occurrence, i.e., if they are reasonably likely to occur or non-reasonably likely to occur, so as to have relevant corrective steps associated with each hazard.

The HACCP plan is formulated as a flow diagram to provide a clear outline of the steps in the production process, which includes steps that occur before and after processing. This plan is approved only after on-site review of the operations for verification of accuracy [41]. Under this system, every procedure and the corresponding obtained data are required for documentation and recordkeeping, which enables the establishment of corrective actions in case of any deviation and verification of the process.

The FSMA was signed on 4 January 2011 with the objective of safeguarding public health by strengthening food safety systems. It gives the FDA new incentives to have the same standards for both imported and domestic food products to build a unified food safety system. It also authorizes preventive controls across the food supply, safety standards for the product, and surveillance of intentional contamination. The law directs the FDA to inspect facilities at regular intervals and to have access to records documenting execution of the food safety plan. It also requires that testing of food products should be done by accredited laboratories to maintain high quality standards. The FSMA also specifies response tools for when problems arise, despite preventive controls, in the form of mandatory recalls and detention of products. The FDA can also suspend the registration of the facility [42,43]. The FSMA proposes seven major regulations which oversee how produce should be grown, packed, processed, shipped and imported. The regulations emphasize consumer-friendly methods for locating food products in case of recall. The regulations also mandate timely registration of manufacturing facilities and transparency for importers to be informed if their food was rejected by another company. Furthermore, they give authority to FDA to detain suspected food items. The FSMA directs a legislative mandate, wherein elaborative, science-based preventive controls are required across the food supply chain [44]

On 17 September 2015, the FDA declared Hazard Analysis and Risk-based Preventive Controls (HARPC) as a new food safety regulation. HARPC comes under FSMA in the FFDCA. It is also called the FSMA food safety plan. The HARPC regulation enforces preventive controls to identify potential risks or threats to the food supply so that proper corrective steps could be taken instead of fixing critical limits, as was the focus in HACCP. The regulation covers aspects such as manufacturing, processing, storage and corresponds to preventive control management components, monitoring procedures, corrective actions and corrections, verification, validation, and preparation of recall plans. The manufacturing facility is required to have a written record of a monitoring program for regular evaluation, which should be approved by the FDA to make sure there are no inadequacies. Corrective actions should be implemented if the preventive controls are not properly executed if there are doubts regarding the effectiveness of the food safety plan and if there are discrepancies in the implementation of a food safety plan. HARPC also requires food facilities to propose steps to confirm that everything is operating correctly, and a record of everything should be documented if ever a reanalysis is required. HARPC makes the manufacturing process more transparent, with improved control over suppliers, partners and buyers [45]. The seven steps of HARPC include: assessment of hazards, including product-specific hazards and facility-specific issues; establishment of preventive controls, including sanitation procedures for food contact points, training for hygiene, monitoring of the environment, and authentication of suppliers; monitoring of the efficacy of the controls; establishment of corrective actions and verification methods; recordkeeping and reanalysis of the plan once every 3 years [46].

The HARPC plan covers food safety concerns beyond CCPs. HARPC relies on aspects such as FDA regulations, standards and guidance documents for preventive control instead of only process controls. All the regulations in FSMA are grounded in HACCP [47,48]. The differences between HACCP and HARPC have been summarized in Table 6.

## 5. HARPC Food Safety Plan for Complementary Food

A food safety plan provides a methodical approach for the minimization of food hazards. It directs the activities required for the safety of the food, including manufacturing, processing and storage of the food product. It describes the preventive controls and the actions required for maintaining food safety. A food safety plan includes process controls, food allergen controls, sanitation controls, supply chain controls and recall plans with corresponding documents for monitoring, corrective action and verification steps [46].

For the safe production of complementary food, the first most important task is developing the product description and understanding its essentials, such as packaging type and material, storage conditions, distribution and intended consumers. Development of a flow diagram gives a detailed understanding of the handling of ingredients and processing steps. It shows the in-process control system spread throughout the different stages of production, from acceptance of raw materials to distribution of the finished product. For example, the growing awareness about the health benefits of ancient seed grains like millets, amaranth arebecoming a popular ingredient in baby food products. Millets have high amounts of protein content, B-group vitamins, minerals such as calcium, iron, phosphorous, manganese and magnesium [49]. The process narrative of complementary food generally adopted by manufacturers is shown in Figure 2 along with the critical control point pertaining to the different hazards [50,51].

### 5.1. Food Processing Steps

#### 5.1.1. Receiving and Storage of Ingredients & Packaging Material

The ingredients and packaging material for the food are received from a supplier verified by the FDA and are stored according to the manufacturer’s requirements. The safety of the ingredients and packaging material is evaluated by Office of Nutritional Product, Labelling and Dietary Supplements (ONLPDS) in consultation with Office of Food Additive Safety (OFAS). The FFDCA act mentions that all manufacturers should use safe food ingredients that are generally recognized as safe (GRAS) or are approved as food additives for use in infant formula. The current laws state that a manufacturer is required to provide the FDA assurance of the nutritional quality of the formulation before marketing [52]. The regulations also require that the manufacturers should screen the products for *Salmonella* and *Cronobacter*, as both are known to thrive in dry places [53]. The ingredients are either shelf-stable, refrigerated, or frozen. Shelf-stable ingredients, such as oils and emulsifiers, are stored at 21 °C; refrigerated ingredients, such as milk, whey protein concentrates and vitamins, are stored at 1.6 °C; and frozen ingredients, such as nucleotides, are kept at −24 °C [54,55]. For thawing of ingredients, the mixture is moved from the freezer to the refrigerator. The packaging material is stored in a dry storage room in the production area.

#### 5.1.2. Processing of Raw Materials

Commercial processing has many stages, with consideration for the type of ingredients, quality and application to ensure the highest quality and safety of the food. The process flow diagram in Figure 2 describes the production of complementary food, adapted to introduce the ancient seed grains amaranth and millets. The other raw materials include proteins, fats, carbohydrates, diluents, minerals, vitamins, and emulsifiers [56]. The production of formula is done mostly by three methods, i.e., wet mixing, dry blending, and a combination of wet and dry mixing. The equipment and utensils for production should be clean, sanitized and made of non-toxic material in a way that does not cause any contamination. The material of the equipment’s construction should not be reactive and absorptive. It also helps in setting and monitoring control parameters and critical limits [57]. The instruments which are used for quality analysis should be accurate, easily read, properly maintained and calibrated before use.

#### 5.1.3. Cleaning and Sorting of Raw Material

Cleaning is a preliminary operation in production and involves the removal of contaminants from the desired raw material. Processing techniques, such as sorting, grading, screening, dehulling, etc., help in obtaining the uniformity of the raw material. The raw materials are sorted into different categories based on their properties, such as size, shape, weight, and color. For size sorting, various kinds of screens and sieves can be utilized [58].

#### 5.1.4. Soaking/Steeping of Grains

The period of soaking of ingredients in water varies depending on the kind of grains used. Traditionally, soaking takes about 8–16 h at temperatures ranging from 10 to 25 °C in a steep tank. In this step, there is an increase in moisture content from 15% to 50% [58].

#### 5.1.5. Fermentation

The fermentation process increases digestibility, palatability and the shelf life of the food product. This process increases the protein digestibility and overall nutritional value of the product. It is performed at 30–50% moisture for 24–72 h at 30 °C using an acid-forming bacteria. Changing the temperature, moisture and type of the inoculum alters the pH of the end product [59].

#### 5.1.6. Wet Milling

Milling helps in reducing the size of the raw material, which lowers the fiber content and bulk of the material. It also lowers the phytates and tannins.. Milling results in a fine particle product but requires equipment that should be used with extensive technical skills for operation. In wet milling, the particles are added in a liquid, where they undergo processes, such as shearing, crushing and attrition. A mill consists of small beads or spheres which are activated by high-speed agitation, which transmits kinetic energy. When the material is pumped inside, the solids suspended in the liquid are torn apart for size reduction [60].

#### 5.1.7. Wet Sieving

This procedure is used to evaluate the size distribution or gradation of the material. It is done to remove fine materials that are difficult to sieve prior to drying. Wet sieving applies to solids that are insoluble in water and remain unchanged, even if heated at 110 °C [61].

#### 5.1.8. Evaporation

This step involves evaporation of the milk mixture. This process is important, as it enhances the spray drying operation and increases the shelf life of the final product. Based on the sensitivity of the ingredients, one-stage or two-stage evaporators are used. Excessive processing can also lead to denaturation of proteins [58].

#### 5.1.9. Spray Drying

The mixture is dried in a spray-dry system. The spray dryer consists of basic elements, such as an air filter, intake fan, heat source, feed source, feed pump, atomizer, drying chamber, cyclone separator, etc. One of the main advantages of a spray drying operation is the short residence time, which makes it suitable for both heat-sensitive and heat-resistant foods. The temperature of the system has to be kept sufficiently high to achieve maximum efficiency [62].

#### 5.1.10. Mixing and Blending

The primary ingredients are required to be blended together. The ingredients are added to the base liquid, which can either be water or skimmed milk, and are stored in a large vessel for complete hydration. The vessels are mostly large stainless-steel tanks in which the base liquid is first added at a temperature of 60 °C, followed by fats, oils and emulsifiers. Vitamin, minerals and stabilizing gums are added later in the process because of their sensitivity to heat. The pasteurization process helps in protecting against spoilage from bacteria, yeast and mold. It involves heating (85–94 °C for 30 s) and cooling of the product under a controlled condition so that microorganisms cannot survive. Several methods can be utilized for pasteurization, of which the most common would be passing the material through a tube adjacent to a plate heat exchanger, so that the mixture is heated indirectly [63].

#### 5.1.11. Standardization

Standardization steps make sure that the key parameters, such as the pH, fat concentration and vitamin and mineral content are in proper amounts. If the materials in the formula are insufficient, then the batch has to be reworked so as to reach the optimum level.

#### 5.1.12. Packaging

Packaging holds a great deal of importance for complementary food, as it separates the food from the external environment and factors like oxygen, light, humidity, temperature and pathogens. Types of packaging and packaging materials can also cause hazards; for example reduced-oxygen packaging can create an environment that is susceptible to growth of *Clostridium botulinum*. A stable and compatible packaging material avoids any kind of physical and chemical damage and extends the shelf life of the product. Depending on the state of the product, packaging is done with materials approved by the FDA. For example, a liquid infant formula is required to be thermally processed in a low-acid package and in a hermetically sealed container. The packaging should be manufactured following the FDA guidance for the preparation of a food contact notification (FCN) for food contact substances (FCS) in contact with infant formula in both liquid and powdered formula. The guidance considers the migration of chemical substances from packaging and other food contact articles.

For migration testing, the data should reflect severe temperature/time conditions using different food simulants and should adhere to prescribed toxicology recommendations [64,65].Common packaging materials utilized for infant formula and complementary food includes metal cans, plastic bags, paper, etc. The high hardness of metal cans contributes to ease of transport and storage, as well as gives anti-extrusion and moisture-proof properties. However, leaching of bisphenol A from the metal can into the formula is one of the major challenges faced by the industry [66]. Single-material pouches made of polypropylene are being used because they help in the recycling process, as this simplifies the process [67]. Overall, packaging materials made of polycarbonate, PVC, polystyrene, glass, polypropylene, low density polyethylene (LDPE), high density polyethylene (HDPE) and polyethylene terephthalate (PET) are considered to be safer alternatives [13]. Sources such as scientific papers, epidemiological studies and data should also be considered in the identification of hazards. The severity of illnesses should be assessed in terms of the probability of the occurrence of a hazard in the absence of preventive controls [68].

#### 5.1.13. Labelling

According to 21 CFR107.10, labelling should be carried out as per Section 201 (z) of the FFDCA. The labelling should include a statement about the number of fluid ounces which provide 100 Kcal and detailed information of the amount of nutrients with prescribed names and units. In addition, a use-by date and directions for use are also required to be added, including aspects such as dilution, storage, etc. [35].

### 5.2. Hazard Analysis

Hazard analysis involves the recognition of hazards associated with the food product or the process. Information from product testing results, consumer complaints, function and design of the facility can be used as an incentive for the analysis. Hazards associated with the raw material can occur because of food allergen cross-contamination or because of pathogens associated with that food product. Many manufacturing processes have a tendency to introduce hazards, such as metal fragments, broken glass, etc. If the parameters of a particular process are not set properly, they can also contaminate the food; for example, improper cooling can lead to growth of microbial pathogens. Different cleaning protocols are required depending on the equipment and how prone the equipment is to damage. Sanitary conditions within the processing facility and employee hygiene are also contributing factors when conducting the hazard analysis. Types of packaging and packaging material can also cause hazards; for example, reduced-oxygen packaging can create an environment that is beneficial for the growth of *Clostridium botulinum*. Sources such as scientific papers, epidemiological studies and data should also be considered in the identification of hazards. The severity of illnesses should be assessed in terms of the probability of the occurrence of a hazard in the absence of preventive controls [69,70].

### 5.3. Monitoring & Corrective Actions

The critical control points are required to be monitored. For infant formula, one of the key control procedures requires analyzing the variance in nutrient levels at all production steps and addresses the situation as appropriate. For analysis, at least one of the indicator nutrients has to be tested during production, after the addition of the nutrient premix and at the final manufacturing step. The analysis must be done in every physical form of the formulation, whether it is powder, ready-to-eat or concentrate. Similar evaluations in complementary food could help children meet adequate nutrition demands.

If monitoring of controls shows any deviation from the critical limits, it is important to take corrective actions. The food safety plan directs the facility regarding whether the corrective action must be enforced immediately, such as through recleaning and sanitizing. It is the responsibility of ONLPDS to evaluate if an infant formula has met the set guidelines [33].

### 5.4. Preventive Controls for Complementary Food

The critical control points associated with complementary food includes supply-chain control, food allergen control, sanitation control and process control. The supply chain control includes verification of controls used by the suppliers to control hazards of raw material before the manufacture receives it. Food allergen controls involve labelling and controls that can prevent cross-contamination, such as product sequencing and sanitation control. Sanitation controls also help to prevent microbial contamination. Process controls include critical parameters, such as time and temperature [46]. The preventive controls required for complementary food production have been summarized in Table 7.

### 5.5. Verification, Validation and Recordkeeping Procedures

Verification includes crosschecking control strategies, such as the calibration of instruments and review of previous records (21 CFR 117.155, 117.160 and 117.165) [38].Validation pertains to referring to scientific literature as a method of handling a hazard. The manufacturer is required to keep records of every single step involved in production in order to monitor the operation, deviations and reasons for failure to meet any required step. In order to avoid any adulteration, a record of all the ingredients (supplier, testing procedure and data, etc.) and packaging materials, including containers and closure, is maintained. The documentation also includes information regarding all equipment, such as software, installation, calibration, testing and validation results.

### 5.6. Types of Recall Associated with Infant Formula/Complementary Food

A recall plan is required to be included in the food safety plan in case a batch of formula or complementary food raises food safety concerns specifically for hazards requiring a preventive control. Information from previous recalls gives the understanding of the likelihood of occurrence. According to 21 CFR 7.3, a recall is classified into 3 categories, I, II and III, wherein class I describes a situation where there is reasonable probability of occurrence of hazard that can cause serious adverse health consequences, class II describes exposure to a violative product which can cause temporary or medically reversible adverse health consequences and class III describes a violative product which is not likely to cause any illness or injury [33]. The written recall plan includes steps to be taken and assigns specific responsibilities to individuals. According to 21 CFR 107.280, manufacturers are required to retain records for at least 1 year after the expiration of the shelf life of the infant formula [35].

### 5.7. Applicability of Regulation and Conduct of Audits

The applicability provision states that if the infant formula manufacturer does not adhere to processing guidelines, quality factors, good manufacturing practices, quality control procedures and quality factor records as stated, the formula is considered to be adulterated. The regulation also covers the registration, submission and notification aspect of the manufactured formula. The audit procedure includes methods that should be used to review if the facility is working in accordance with current good manufacturing practices with quality control procedures which are necessary for infant formula. The audit plan includes an evaluation of production and in-process control systems in comparison with the written procedures, review of the records and deviation at every stage necessary to avoid adulteration.

The U.S Federal Food Law emphasizes that any adulterated or misbranded food cannot be sold. Adulterated food consists of food with any substance, additive, pesticide, or putrid and chemical residue that is injurious to human health. It also includes food which is prepared under unsanitary conditions, or food products made from a diseased animal. Misbranded food consists of food whose labelling or advertising is false and misleading. It also restricts the selling of food under another name and including the word “imitation” when required. The information of the label should be accurate in terms of weight, measure or numerical count [46].

## 6. Impact of COVID-19

Although globalization has improved food systems in terms of accessibility, availability, and affordability, it has also made the food supply chain vulnerable to outbreaks. The COVID-19 pandemic, instigated by severe acute respiratory syndrome coronavirus 2 (SARS-CoV-2), has affected almost 188 countries, leading to the death of more than 1 million people thus far [71]. The virus spreads by close contact from small droplets generated via coughing, sneezing, and talking. COVID-19 can be contracted by a person while touching a surface or object, including food or food packaging, and then touching their own mouth, nose or eyes [72]. The COVID-19 pandemic has caused a major disruption in social and economic systems, wherein the food processing industry is currently facing a widespread supply shortage. Food processing facilities are required to follow COVID-19 protocols that are set by local and state governments depending on the community spread of SARS-CoV-2. At present, the virus can be prevented by washing hands with soap and water or sanitizing with 60% alcohol, which is a common Good Manufacturing Practice (GMP) followed for food safety in food processing plants [73]. Although foodborne exposure to SARS-CoV-2 is uncertain as a route of transmission, there have been guidelines issued by the Centers for Disease Control and Prevention (CDC) accounting for food safety in the kitchen, handling of packaged food, bulk meat and poultry and other foods [74]. For handling COVID-19 contamination in a food facility, the Codex Alimentarius commission has established several global standards to control viruses in food specifically in terms of food hygiene. These standards include procedures for general principles in food hygiene, food import and export inspection codes, guidelines for design, operation, assessment, accreditation, etc. Codex texts are benchmarks and framework for food safety in the global food trade. The COVID-19 pandemic has disrupted the schedule of Codex sessions, but the regulatory body is exploring different ways to maintain the momentum of standard setting work through their electronic working groups (EWGs) and by virtual meetings [75]. To deal with the COVID-19 crisis, the ISO has compiled a list of standards to support global efforts. These standards include regulations with respect to clothing for protection against infectious agents, such as medical face masks, protective gloves, and guidelines for supporting a vulnerable person in an emergency. The ISO has also listed standards for ensuring personal protection, personal hygiene and monitoring of medical equipment, such as anesthetic and respiratory equipment, biological evaluation of medical devices and their quality management, biocompatibility evaluation of breathing gas pathways, requirements for the evaluation of the performance of quantification methods for nucleic acid target sequences (qPCR and dPCR), etc. [76]. The CDC, FDA and USDA Food Safety and Inspection Service (FSIS), with a full-time staff of FDA’s Coordinated Outbreak Response and Evaluation (CORE) network, have continued their operations for monitoring and tracking any outbreak situation [77]. Due to the COVID-19 crisis, several domestic and foreign food safety surveillance inspections have either decreased or been postponed. Food safety trainings have also decreased, and the mode of instruction has shifted from face-to-face to a remote online training mode.

## 7. Future Potentials

Adequate and appropriate macro- and micronutrients are one of the most important factors which sets the bifacial relationship between infection and immunity. Around the world, food regulations are being enforced to mandate industries to adopt best food handling practices to avoid any risk of foodborne illness. Even with the presence of a well-crafted system, there are no strict regulations for complementary food specifically in terms of nutritional requirements. These regulations can help children ranging in age from 6 months to 24 months meet nutritional goals for their optimal growth and development. Currently, undernutrition is responsible for the death of 45% of children under the age of five, and, hence, it is crucial to take steps to prevent it. National food laws and regulations constitute the food safety model, which deals with food hazards through response actions and effective plans for mitigation of risk. Food safety regulations are based on comprehensive risk analyses performed by the regulatory bodies. These regulatory bodies have a unified mission to evaluate and inspect implementation and are ideally organized in a way wherein the state and local government work as an integrated enterprise with well-defined responsibilities and adequate funding. The Baby Food Safety Act of 2021, which was introduced on March 25th, 2021, is one such example of how safety requirements can be imposed, as the bill establishes the permissible levels of cadmium (5 ppb), lead (5 ppb), mercury (2 ppb) and inorganic arsenic (10 ppb) in infant and complementary food products. The act is an amendment to the Federal Food, Drug and Cosmetic Act (21 U.S.C 321) and states that preventive control measures should be employed.

A strong food safety system should always be kept modernized and updated with the ongoing settings. Earlier, the food safety culture was limited to addressing how the people involved in production should work to make a product safe, but now it is considered that the safety of both the product and people deserve equal attention. The unprecedented situation of the COVID-19 pandemic unveiled the requirement of a digital and transparent food system, where issues such as control parameters and contamination are also needed to be taken care of. It can be further improved by incorporating technological changes, such as blockchain technology, which helps enable the distribution of a database across a network of computers, helping in traceability across the processing chain. The data generated from improved AI/ machine learning technologies for food safety and process controls can help in performing predictive analytics to prevent contamination at any step along the supply chain.

## Figures and Tables

**Figure 1 foods-10-02199-f001:**
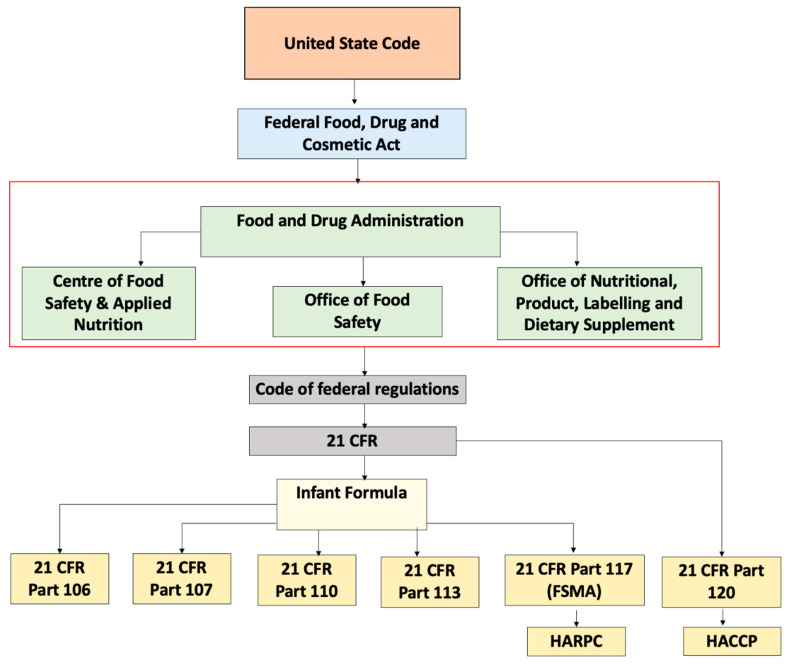
Schematic representation of regulations governing infant formula in United States.

**Figure 2 foods-10-02199-f002:**
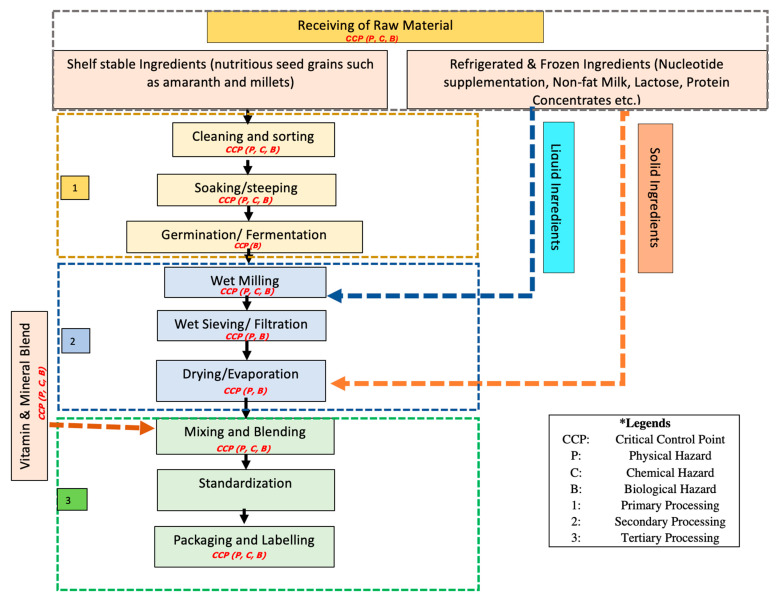
Process flow diagram of complementary food illustrating all the processing steps involved in production, wherein different controls can be set for monitoring the critical limits.

**Table 1 foods-10-02199-t001:** Different stages of child development and type of food required [3].

Category	Age Group	Food
Neonate	Birth to 28 days	Breast Milk/Infant formula
Infant	1 month–6 months	Breast Milk/Infant formula
	6 months–24 months	Complementary Food
Children	2 to 12 years	Standard food
Adolescent	12 to 16 years	Standard food

**Table 2 foods-10-02199-t002:** List of recent infant formula and complementary food recall outbreaks and the possible causes [28,29,30].

Name of Company and Product	Date of Recall	Reason for Recall/ Outbreak
Beech-Nut Nutrition	9 June 2021	High arsenic levels
Calcilo Formula	17 September 2019	Inconsistency in aroma and color
Mountain Mel’s Peaceful Baby Herbal Tea	29 August 2019	Contamination with *Salmonella*
Heinz Turkey Stew Baby Food	December 2019	Possibility of being contaminated with insects
Parent’s Choice Advantage Infant Formula	21 June 2019	Potential presence of metal foreign matter
Lactalis Formula	11 December 2017	Contamination with *Salmonella*
PC Organic Baby Food	7 February 2017	Growth of *Clostridium botulinum*
H-E-B Baby Food	18 November 2016	Small piece of rubber found in one product

**Table 3 foods-10-02199-t003:** Essential nutrients listed in Codex standards for follow up formula (complementary food) with minimum and maximum contents. Obtained from *Codex Alimentarius Standard for Follow Up Formula* [11].

Essential Nutrient	Minimum Content per 100 Available Calories	Maximum Content per 100 Available Calories
Protein	3 g with nutritional with quality equivalent to or greater than casein.	5.5 g
Fat	3 g (linolenic acid should not be less than 300 mg per 100 calories)	6 g
Carbohydrates	Nutritionally available carbohydrates added in accordance to required energy density	-
Vitamins		
Vitamin A	167.5 mg	502.5 mg
Vitamin D	26.8 mg	80.4 mg
Vitamin C (Ascorbic acid)	8 mg	-
Vitamin B1 (Thiamine)	40 µg	-
Vitamin B2	60 µg	-
Nicotinamide	250 µg	-
Vitamin B6	45 µg	-
Folic Acid	4 µg	-
Pantothenic acid	300 µg	-
Vitamin B12	0.15 µg	-
Vitamin K1	4 µg	-
Biotin	1.5 µg	-
Minerals		
Sodium	20 mg	85 mg
Potassium	80 mg	-
Chloride	55 mg	-
Calcium	90 mg	-
Phosphorous	60 mg	-
Magnesium	6 mg	-
Iron	1 mg	2 mg
Iodine	5 µg	-
Zinc	0.5 mg	-

**Table 4 foods-10-02199-t004:** Examples of private food safety standards for certification for food industries [32].

Global Safety Standard Certification	Description
International Organization for Standardization (ISO 22000)	The ISO 22000 standard describes the food safety management system requirements for any organization involved in the food chain, such as ingredient producers, retailers, catering services, transportation, etc. Any organization can pursue certification and registration if it conforms with this standard.
Safe Quality Food (SQF) 1000/2000	It is a Hazard Analysis and Critical Control Points (HACCP)-based certification system for food safety and quality of ingredients, packaging, farming, packing houses, etc.
British Retail Consortium (BRCGS)	This standard has been developed in collaboration with the industry for provision of product safety and quality.
PrimusGFS	This certification program is a farm-focused Global Food Safety Initiative (GFSI).
Global Good Agricultural Practices (GAP)	This certification program covers primarily agricultural crops, such as fruits, vegetables, hops, tea, etc.
International Featured Standards (IFS)	This certification program covers the processes in the supply chain by doing risk-based assessments.
Food Safety Management Certification (FSSC) 22000	This certification is based on ISO 22000, ISO 9001, ISO/TS22003 and ISO 22003 and confirms food safety and quality of the organization certified.

**Table 5 foods-10-02199-t005:** Code of Federal Regulations (CFR) for the manufacturing of infant formula [34,35,36,37,38].

Regulation	Description
21 CFR Part 106	Current Good Manufacturing Practices (cGMP), production and in process control system, prevention of adulteration by workers, facilities, equipment or utensils, ingredients, microorganism, packaging and labelling, audit of cGMP, record keeping, registration, submission and notification.
21 CFR Part 107	General provision and applicability, labelling, exemptions, nutrient requirements, recalls, elements of infant formula recall, notification requirements and record retention.
21 CFR Part 110	cGMP, packaging, holding of human food, production and process control.
21 CFR Part 113	Thermally processed low acid food packaged in hermetically sealed containers.
21 CFR Part 117	cGMP, hazard analysis and risk-based preventive controls for human food, requirements applying to records that must be established and maintained and the supply chain program
21 CFR Part 120	HACCP, The regulation primarily pertains to juice products.

**Table 6 foods-10-02199-t006:** Differences between HACCP and HARPC in terms of various elements of food safety plans [44,46].

Hazard Analysis and Critical Control Point (HACCP)	Hazard Analysis and Risk-Based Preventive Controls (HARPC)
HACCP is based on the Codex Alimentarius and guidelines given by the National Advisory Committee on Microbiological Criteria for Food. It is required by meat, poultry, seafood and juice industries.	HARPC is a food safety plan based on the Food Safety Modernization Act (FSMA).
HACCP only covers chemical, biological and physical hazards.	In addition to chemical, biological and physical hazards, HARPC also considers radiological hazards, natural toxins, pesticides and drug residues, parasites, allergens and unapproved food and color additives, and non-intentional and intentional economically motivated hazards.
Critical control points are required for the processes.	Critical control points are required for processes and other points as required for the food safety. (21CFR117.135)
According to HACCP plan, the critical limits are set at CCPs.	HARPC also has a set of parameters and values in terms of maximum and minimum values. Optimization of these values minimizes the occurrence of hazards. (21CFR117.135 c (1))
HACCP requires all the set process controls to be verified.	In HARPC, verification is required for all preventive and process controls. Supplier verification is also required.
Recall is not required in the plan.	According to HARPC, a written recall plan is mandatory when a hazard is identified. The written plan should include procedures which explains the aftermath steps and relevant responsibility.
The HACCP is required to be reviewed at least once a year or when required.	The HARPC plan can be reviewed once every three years or when required.

**Table 7 foods-10-02199-t007:** Potential hazards associated with different processing steps of production of complementary food.

Source	Potential Hazards	Quality Procedure & Preventive Control
Receiving raw material (ingredients, packaging and labelling material)	Biological because of growth of pathogens and chemical because of possible allergen cross-contact	Allergen preventive control, supply chain preventive control
Storage of raw material	Chemical hazard because of possible oxidative rancidity	Sanitation preventive control
Weighing and mixing of ingredients	Biological hazards because of environmental pathogens and physical hazards because of chances of metal inclusion from metal-metal contact during mixing	Sanitation and process preventive control
Processing (homogenization, evaporation and spray drying)	Biological hazards can occur if the set temperature is not reached, and physical hazards can occur if there is metal to metal contact during processing	Process preventive control
Packaging	Chemical hazard because of allergen cross-contact	Allergen preventive control
Final Product	Biological because of growth of pathogens and chemical hazards because of possible allergen cross-contact	Allergen preventive control, supply chain preventive control
Storage of final product	Chemical hazard because of possible oxidative rancidity	Sanitation preventive control

## Data Availability

The data supporting the results of this study are available from the corresponding author, Kiruba Krishnaswamy (krishnaswamyk@umsystem.edu).

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
