# Peer review of "Hazard Analysis and Risk-Based Preventive Controls (HARPC): Current Food Safety and Quality Standards for Complementary Foods"

_foods, 2021, doi:10.3390/foods10092199_

Round 1
Reviewer 1 Report
Title: The current title does not reflect the content of the manuscript. I’d suggest the following:
Hazard analysis and risk-based preventive controls (HARPC) plan and current food safety and quality standards for complementary foods
- Microorganisms and foodborne illnesses….
Suggest adding other chemical and physical hazards and associated chemical foodborne illnesses to this section. Add in the melamine scandal (since one of your keywords is food fraud) too. Authors could also move Table 5 to this section and link the hazards / foodborne illnesses to the recall.
- HARPC plan
It is a difficult balance when deciding whether to provide a generic HARPC food safety plan vs. a specific one. To Foods’ readers, a specific plan would be more useful, and this would also meet the aims of your manuscript. Rather than broad brushing the generic processes involved in producing complementary food, select one specific type of complementary food commonly available in U.S. For example, let’s say the cereal is made from hydrolysed wheat, wheat, rice, corn, whole grain, oats and vitamins and mineral blends. Use this to develop your process flow diagram and HARPC plan.
Section 5.1 Food processing steps would need to be revised to reflect a more specific process flow.
5.2 Hazard analysis should be more specific and based the assessment on the type of raw materials and ingredients used. Please include the CCPs in the process flow diagram once identified.
Include a section on preventive controls for sanitation, training, environment monitoring etc.
Please separate 5.3 Monitoring and Corrective actions and discuss this specifically with regards to the controls identified in ‘Preventive controls’ and what are the specific correction action measures.
Author Response
We would like to thank you providing valuable feedback. Kindly see our reply to the comments.
- Title: The current title does not reflect the content of the manuscript. I’d suggest the following: Hazard analysis and risk-based preventive controls (HARPC) plan and current food safety and quality standards for complementary foods
Response: Thank you so much for your thoughtful suggestion. We have changed the title of the manuscript accordingly.
- Microorganisms and foodborne illnesses…. Suggest adding other chemical and physical hazards and associated chemical foodborne illnesses to this section. Add in the melamine scandal (since one of your keywords is food fraud) too. Authors could also move Table 5 to this section and link the hazards / foodborne illnesses to the recall.
Response: Thank you so much for the suggestion. We have added the following lines for including chemical and physical hazards. The recall/outbreak table is also moved up in this section.
Line 93-97
“Foodborne illness is usually associated with the heat labile micronutrients that are added after the heating step to keep the nutritional content compatible with standards set by regulatory agencies (Anselmo et al., 2019). The major reasons for an outbreak are clas-sified into three categories biological, chemical, and physical hazards (FDA, 2018b).”
Line 105-106
“Microorganism such as bacteria, viruses, yeast etc. are the major cause of biological hazards.”
Line 135-163
“Presence of any chemical residue such as pesticides, antibiotics, herbicides, insecticides, sanitizers etc., in food can lead to chemical hazard. These chemical residues should either be in permissible amount that is set by the regulatory authorities or should not be found in the food product entirely. The chemical hazards also include some naturally occurring chemicals such as mycotoxins. The sources of chemical hazards include environmental contamination, processing steps, additives etc. In case of baby food, heavy metals such as cadmium, lead, arsenic, and mercury are major cause of concern. Exposure to these heavy metals can lead to severe health issues such as brain damage, lung, and skin cancer etc.
In 2008, 300,000 people reported illness wherein 860 children were hospitalized out of which 154 were classified as severe cases due to consumption of substandard milk and infant formula leading to kidney stone formation and kidney failures. The reason for the outbreak was melamine which was added to make appear that the product is higher in protein content because of its high nitrogen content. Melamine is not approved by food standard commissions. The outbreak prompted authorities to improve the safety standards wherein even low levels of chemicals should be considered harmful (Branigan, 2008).
Physical hazards include discovery of foreign objects such as broken glass, plastic, stones, wood, or metal in the food product. The sources for physical hazards majorly include the glass storage containers, packaging material, equipment etc. Children are more prone to choking hazards. The physical hazards can lead to cut in throat and can also affect intestines. Table 2 summarizes some of the recent recalls that occurred because of these hazards.”
Line 165-167
Table 2. List of recent infant formula and complementary food recall and the possible causes (Corley, 2020), (Rowan, 2020).
Name of Company and Product |
Date of Recall |
Reason for Recall/ Outbreak |
Beech-Nut Nutrition |
June 9, 2021 |
High Arsenic Levels |
Calcilo Formula |
Sept 17, 2019 |
Inconsistency in aroma and color. |
Mountain Mel’s Peaceful Baby Herbal Tea |
August 29, 2019 |
Contamination with Salmonella. |
Heinz Turkey Stew Baby Food |
December 2019 |
Possibility of being contaminated with insects. |
Parent’s Choice Advantage Infant formula |
June 21, 2019 |
Potential presence of metal foreign matter |
Lactalis Formula |
December 11, 2017 |
Contamination with Salmonella. |
PC Organic Baby Food |
February 07, 2017 |
Growth of Clostridium botulinum |
H-E-B Baby Food |
November 18, 2016 |
Small piece of rubber found in one product |
- It is a difficult balance when deciding whether to provide a generic HARPC food safety plan vs. a specific one. To Foods’ readers, a specific plan would be more useful, and this would also meet the aims of your manuscript. Rather than broad brushing the generic processes involved in producing complementary food, select one specific type of complementary food commonly available in U.S. For example, let’s say the cereal is made from hydrolysed wheat, wheat, rice, corn, whole grain, oats and vitamins and mineral blends. Use this to develop your process flow diagram and HARPC plan.
Response: Thank you so much for the feedback. We have changed the generic process flow diagram into a process flow diagram for production of a millet based complementary food. Following lines indicate the changes:
Line 337- 342
“With the growing awareness about health benefits of millets, it is becoming a popular ingredient in baby food products. Millets have high amount of protein content, B-group vitamins, minerals such as calcium, iron, phosphorous, manganese and magnesium.”
- Section 5.1 Food processing steps would need to be revised to reflect a more specific process flow.
Response: Thank you so much. We have edited the process flow diagram and accordingly edited the processing steps. Line 343
Line 351-530
“5.1. Food processing steps
5.1.1. Receiving and storage of ingredients & packaging material: The ingredients and packaging material for the food are received from a supplier verified by the FDA and are stored according to manufacturer’s requirements. The safety of ingredients and pack-aging material is evaluated by ONLPDS in consultation with OFAS. The FFDCA act mentions that all manufacturers should use safe food ingredients that are generally recognized as safe (GRAS) or are approved as food additives for use in infant formula. The current laws state that a manufacturer is required to provide the FDA assurance of the nutritional quality of the formulation before marketing (FDA,2020d). The regulations also require that the manufacturers should screen the products for Salmonella and Cronobacter as both of them are known to thrive in dry places (Connor, 2014). The ingredients are either shelf stable, refrigerated or frozen. The shelf stable ingredients like oils and emulsifiers are stored at 21°C, refrigerated ingredients such as milk, whey protein con-centrates, vitamins, et are stored at 1.6 °C and frozen ingredients such as nucleotides are kept at -24 °C (Mead Johnson Nutrition, 2020; Similac, 2020). For thawing of ingredients, the mixture is moved from freezer to refrigerator. The packaging material is stored in the dry storage room in the production area.
5.1.2. Processing of Raw Materials: The commercial processing has many stages with consideration for type of ingredients, quality and the application to ensure the highest quality and safety of the food. Figure 2 describes a general process flow diagram for production of complementary food for a seed grain based raw material such as amaranth and millets. The other raw materials include proteins, fats, carbohydrates, diluents, minerals, vitamins, and emulsifiers (Hejazi, 2016) (Institute of Medicine (US) and N.R.C (US) Committee, 1998) The production of the formula is done mostly by three methods i.e., wet mixing, dry blending, and combination of wet and dry mixing. The equipment and utensils for production should be clean, sanitized and made of non-toxic material in a way that does not cause any contamination. The material of equipment construction should not be reactive and absorptive. It also helps in setting and monitoring control parameters and critical limits (Almeida et al., 1999). The instruments which are used for quality analysis should be accurate, easily read, properly maintained and calibrated before use.
5.1.3. Cleaning and Sorting of Raw Material: Cleaning is a preliminary operation in the production and involves removal of contaminants from the desired raw material. Pro-cessing techniques, such as sorting, grading, screening, dehulling etc., help in obtaining the uniformity of the raw material. The raw material is sorted into different categories based on their properties, such as size, shape, weight, and color. For size sorting, various kinds of screens and sieves can be utilized.
5.1.4. Soaking/Steeping of Grains: Depending on the kind of cereal, soaking is done by putting the ingredients in water for a defined period of time. Traditionally, it is done for 8-16 hours at temperatures ranging from 10 to 25 °C in a steep tank. In this step, there is an increase in moisture content from 15% to 50% (Control & N.R.C (US) Subcommittee, 1992).
5.1.5. Fermentation: A fermentation process increases digestibility, palatability and shelf life of the food product. This process increases the protein digestibility and overall nu-tritional value of the product. It is performed at 30-50% moisture for 24-72 hours at 30 °C using an acid forming bacteria. Changing the temperature, moisture and type of inoc-ulum alters the pH of the end product (Verni et al., 2019)
5.1.6. Wet Milling: Milling helps in reducing the size of the raw material which lowers the fiber content and bulk of the material. It also lowers the phytates and tannins (Control &N.R.C (US) Subcommittee, 1992). Milling results in a fine particle product but requires equipment that should be used with extensive technical skills for operation. In wet milling, the particles are added in a liquid where they undergo processes like shearing, crushing and attrition. A mill consists of small beads or spheres which are activated by high-speed agitation that transmits the kinetic energy. When the material is pumped inside, the solids suspended in a liquids are torn apart for size reduction (Rosentrater & Evers, 2018).
5.1.7. Wet Sieving: This procedure is used to evaluate the size distribution or gradation of the material. It is done to remove fine materials that are difficult to sieve prior to drying. Wet sieving applies to solids that are insoluble in water and remain unchanged, even if heated at 110 °C (Daeschner, 1969).
5.1.8. Evaporation: This step involves evaporation of the milk mixture. This process is important as it enhances the spray drying operation and increases the shelf life of the final product. Based on the sensitivity of the ingredients, a one stage or two stage evaporators are used. Excessive processing can also lead to denaturation of proteins (Control & N.R.C (US) Subcommittee, 1992).
5.1.9. Spray Drying: The mixture is dried in a spray-dry system. The spray dryer consists of basic elements, such as air filter, intake fan, heat source, feed source, feed pump, atomizer, drying chamber, cyclone separator etc. One of the main advantages of a spray drying operation is the short residence time which makes it suitable for both heat sensitive and heat resistant foods. The temperature of the system has to be kept sufficiently high to achieve maximum efficiency (Gharsallaoui et al., 2007).
5.1.10. Mixing and Blending: The primary ingredients are required to be blended to-gether. The ingredients are added to the base liquid which can either be water or skimmed milk and are stored in a large vessel for complete hydration. The vessels are mostly large stainless-steel tanks in which the base liquid is first added at temperature 60 °C, followed by fats, oils and emulsifiers. Vitamin, minerals and stabilizing gums are added later in the process because of their sensitivity to heat. Pasteurization process helps in protecting against spoilage from bacteria, yeast and mold. It involves heating (85-94 0C for 30 s) and cooling of the product under a controlled condition so that microorganisms don’t survive. Several methods can be utilized for pasteurization of which the most common one would be passing the material through a tube adjacent to a plate heat ex-changer so that the mixture gets heated indirectly (Dali et al., 2017).
5.1.11. Standardization: Standardization steps make sure that the key parameters, such as pH, fat concentration and vitamin and mineral are in proper amount. If the material in the formula are insufficient, then the batch has to be reworked so as to reach optimum level.
5.1.12. Packaging: Packaging holds a great importance for the complementary food as it separates the food from external environment and factors like oxygen, light, humidity, temperature and pathogens. Types of packaging and packaging material can also cause hazards, for example reduced oxygen packaging can create an environment that is susceptible for growth of Clostridium botulinum. A stable and compatible packaging material avoids any kind of physical and chemical damage, as well as extends the shelf life of the product. Depending on the state of the product, packaging is done with ma-terials approved by FDA. For example, for a liquid infant formula is required to be thermally processed in a low acid package and in a hermetically sealed container. The packaging should be manufactured following the FDA guidance for preparation of Food Contact Notification (FCN) for Food Contact Substances (FCS) in contact with infant formula in both liquid and powdered formula. The guidance contemplates the migration of chemical substances from packaging and other food contact articles.
For migration testing, the data should reflect severe temperature/time conditions using different food simulants and should adhere to prescribed toxicology recommen-dations (Jablonski et al., 2019)(Office of Food Additive Safety, 2019). The common packaging material utilized for infant formula and complementary food includes metal cans, plastic bags, paper etc. The high hardness of metal cans leads to ease of transport and storage, as well as gives anti-extrusion and moisture proof properties. However, leaching of bisphenol A from the metal can into the formula is one of the major challenges faced by the industry (Environmental Working Group, 2007). Single material pouches made of polypropylene are being used because they help in the recycling process as this simplifies the process (Lingle, 2020). Overall, packaging materials made of polycarbonate, PVC, polystyrene, glass, polypropylene, LDPE, HDPE and PET are considered to be safer alternatives (Clear and Well, 2016). Sources such as scientific papers, epidemiological studies and data should also be considered in identification of hazards. The severity of illnesses should be assessed in terms of the probability of occurrence of a hazard in the absence of preventive control(FDA, 2018a).
5.1.13. Labelling: According to 21 CFR107.10, the labelling should be done as per section 201 (z) of FFDCA. The labelling should include a statement about number of fluid ounces which provide 100 Kcal and detailed information of the amount of nutrients with pre-scribed names and units. In addition, a use-by date and directions for use are also required to be added including aspects such as dilution, storage etc., (FDA, 2019a).”
- Hazard analysis should be more specific and based the assessment on the type of raw materials and ingredients used. Please include the CCPs in the process flow diagram once identified. Include a section on preventive controls for sanitation, training, environment monitoring etc. Please separate 5.3 Monitoring and Corrective actions and discuss this specifically with regards to the controls identified in ‘Preventive controls’ and what are the specific correction action measures.
Response: Thank you so much for your valuable feedback. We have added the CCPS in the process flow diagrams and edited the hazard analysis section. The preventive control section has also been added accordingly. The following lines indicate the changes.
Line 531-547
“5.2 Hazard Analysis
Hazard analysis involves recognition of hazards associated with the food product or the process. Information from product testing results, consumer complaints, function and design of the facility can be used as an incentive for the analysis. Hazards associated with the raw material can occur because of food allergen cross-contamination or because of pathogens associated with that food product. Many of the manufacturing processes have a tendency to introduce hazards, such as metal fragments, broken glass, etc. If the pa-rameters of a particular process are not set properly, they can also contaminate the food, for example, improper cooling can lead to growth of microbial pathogens. Different cleaning protocols are required to be made depending on the equipment and how prone they are to damage. Sanitary conditions within the processing facility and employee hygiene are also contributing factors while conducting the hazard analysis. Types of packaging and packaging material can also cause hazards, for example reduced oxygen packaging can create an environment that is susceptible for growth of Clostridium botulinum. Sources such as scientific papers, epidemiological studies and data should also be considered in identification of hazards. The severity of illnesses should be assessed in terms of the probability of occurrence of a hazard in the absence of preventive control (FDA, 2018a).”
Line 601- 610
“5.4. Preventive Controls for Complementary Food
The critical control points associated with complementary food includes supply-chain control, food allergen control, sanitation control and process control. The supply chain control includes verification of controls used by the suppliers to control hazards of raw material before the manufacture receives it. Food allergen controls involve labelling and controls that can prevent cross-contamination, such as product sequencing and sanitation control. Sanitation controls also helps to prevent microbial contamination. The process controls include critical parameters, such as time and temperature (FDA, 2018a).”
Line 612-613
Table 7. Potential hazards associated with different processing step of production of complementary food.
Source |
Potential hazards |
Quality Procedure & Preventive control |
Receiving raw material (ingredients, packaging and labelling material) |
Biological because of growth of pathogens and chemical hazard because of possible allergen cross contact. |
Allergen preventive control, supply chain preventive control |
Storage of raw material. |
Chemical hazard because of possible oxidative rancidity. |
Sanitation preventive control. |
Weighing and mixing of ingredients |
Biological hazard because of environmental pathogens and physical hazard because of chances of metal inclusion from metal-metal contact during mixing. |
Sanitation and process preventive control. |
Processing (Homogenization, evaporation and spray drying) |
Biological hazard can occur if the set temperature is not reached. Physical hazard can occur if there is metal to metal contact during mixing |
Process preventive control. |
Packaging |
Chemical hazard because of allergen cross contact |
Allergen preventive control |
Receiving raw material (ingredients, packaging and labelling material) |
Biological because of growth of pathogens and chemical hazard because of possible allergen cross contact. |
Allergen preventive control, supply chain preventive control |
Storage of raw material. |
Chemical hazard because of possible oxidative rancidity. |
Sanitation preventive control. |
Reviewer 2 Report
Thank you for allowing me to review this manuscript on food safety and infant nutrition. Infants are an highly vulnerable population, so this paper is of great importance and appropriate for this journal. I'm not an expert in infant nutrition, but am aware of food safety regulations around the world. Goals were clearly presented in the outset. This manuscript does a good job in reviewing what has happening in the U.S. over the last decade, and how the country's emphasis has changed on food safety. It would be appropriate to see how things can improve even more given new risks. In my view the paper is not critical enough about how we can improve policies and oversight in the U.S., in relation to baby foods in general. What are implications for the future? what needs to change given what we know? Is industry collaborative? What are the new potential threats on the horizon? Not one mention of COVID which I find quite surprising. Something needs to be mentioned. How will parents' perception of risks be impacted by COVID when dealing with infant foods?
I really like how the paper conceptually presents the value chain and the different elements of food safety policy in the U.S.
One question though. Table 5 shows a list of recent recalls, but the newest one is 2 years old. We have not seen a recall since? If so, it would worthy of an explanation.
Author Response
We would like to thank you providing valuable feedback. Kindly see our reply to the comments.
- Thank you for allowing me to review this manuscript on food safety and infant nutrition. Infants are a highly vulnerable population, so this paper is of great importance and appropriate for this journal. I'm not an expert in infant nutrition but am aware of food safety regulations around the world. Goals were clearly presented in the outset. This manuscript does a good job in reviewing what has happening in the U.S. over the last decade, and how the country's emphasis has changed on food safety. It would be appropriate to see how things can improve even more given new risks. In my view the paper is not critical enough about how we can improve policies and oversight in the U.S., in relation to baby foods in general. What are implications for the future? what needs to change given what we know? Is industry collaborative? What are the new potential threats on the horizon?
Response: Thank you so much for your insightful and thoughtful feedback. We have addressed the questions in the section future potentials:
Line 707-742
“Future Potentials
Adequate and appropriate macro/micronutrients are one of the most important factors which sets the bifacial relationship between infection and immunity. Around the world, food regulations are being enforced to mandate industries to adopt best food handling practices to avoid any risk of foodborne illness. Even with the presence of a well-crafted system, there are no strict regulations for complementary food specifically in terms of nutritional requirement. These regulations can help children in the age from 6 months to 24 months meet nutritional goals for their optimal growth and development. Currently, undernutrition is responsible for death of 45% of children under the age of five and hence, it is crucial to take steps to prevent it. National food laws and regulations constitute the food safety model which deals with the food hazards with a response action and effective plan for mitigation of risk. The food safety regulations are based on com-prehensible risk analysis done by the regulatory bodies. These regulatory bodies have a unified mission to evaluate and inspect implementation and are ideally organized in a way wherein the state and local government work as an integrated enterprise with well-defined responsibilities and adequate funding. The Baby Food Safety Act of 2021 that was introduced on March 25th , 2021 is one such example how the safety requirements can be imposed as the bill establishes the permissible levels of cadmium (5 ppb), lead (5 ppb), mercury (2 ppb) and inorganic arsenic (10 ppb) in the infant and complementary food products. The act is an amendment to Federal Food, Drug and Cosmetic Act (21 U.S.C 321) and states that preventive control measures should be employed.
A strong food safety system should always be kept modernized and updated with the ongoing settings. It can be further improved by incorporating technological changes, such as blockchain technology which helps enable distribution of a database across a network of computers helping in traceability across the processing chain. The data generated from this process can help in performing predictive analytics for any possible contamination at any step. The unprecedented situation of the COVID-19 pandemic unveiled the requirement of a digital and transparent food system where issues such as control parameters and contamination are also needed to be taken care off. Earlier, the food safety culture was limited to addressing how the people involved in production should work to make a product safe, but now it is being considered that safety of both product and people deserves equal attention.”
- Not one mention of COVID which I find quite surprising. Something needs to be mentioned. How will parents' perception of risks be impacted by COVID when dealing with infant foods?
Response: Thank you so much for this suggestion. We have added one new section in the manuscript to address the issue that extensively talks about COVID-19 and how different regulatory agencies changed their policies.
Line 659-706
“6. Impact of COVID-19
Globalization, while improving food system in terms of accessibility, availability and affordability, has also made the food supply chain vulnerable to outbreaks. The COVID-19 pandemic instigated by severe acute respiratory syndrome coronavirus 2 (SARS-CoV-2) affected almost 188 countries leading to the death of more than 1 million people this far (WHO, 2020 a). The virus spreads by close contact from small droplets generated via coughing, sneezing and talking. COVID-19 can be contracted by a person, while touching a surface or object, including food or food packaging and then touching their own mouth, nose and eyes (WHO, 2020b). The COVID-19 pandemic has caused a major disruption in the social and economic systems wherein the food processing industry is currently facing a widespread supply shortage. Food processing facilities are required to follow COVID-19 protocols that are set by local and state governments depending on the community spread of SARS-CoV-2. At present, the virus can be prevented by washing hands with soap and water or sanitizing with 60% alcohol, which is a common Good Manufacturing Practice (GMP) followed for food safety in food processing plants (WHO, 2020c). Although foodborne exposure to SARS-CoV-2 is uncertain as a route of trans-mission, there have been guidelines issued by the Centers for Disease Control and Prevention (CDC) accounting for food safety in the kitchen, handling of packaged food, bulk meat and poultry and other foods (CDC, 2020). For handling Covid-19 contam-ination in a food facility, the Codex Alimentarius commission has established several global standards to control viruses in food specifically in terms of food hygiene. These standards include procedures for general principles in food hygiene, food import and export inspection codes, guidelines for design, operation, assessment, accreditation, etc. Codex texts are benchmarks and framework for food safety in global food trade. The Covid-19 pandemic has disrupted the schedule of Codex sessions, but the regulatory body is exploring different ways to maintain the momentum of standard setting work through their electronic working groups (EWGs) and by virtual meetings (FAO/WHO, 2020). To deal with the Covid-19 crisis, ISO has compiled a list of standards to support the global efforts. These standards include regulations with respect to clothing for protection against infectious agents, such as medical face mask, protective gloves, and guidelines for supporting a vulnerable person in an emergency. ISO has also listed standards for en-suring personal protection, personal hygiene and monitoring of medical equipment, such as anesthetic and respiratory equipment, biological evaluation of medical devices and its quality management, biocompatibility evaluation of breathing gas pathways, re-quirements for evaluation of performance of quantification method for nucleic acid target sequences- qPCR and dPCR etc. (ISO, 2020). Due to the COVID-19 crisis, several domestic and foreign food safety surveillance inspections have either decreased or been postponed. Food safety trainings have also decreased, wherein the mode of instruction has shifted from face-to-face to a remote training mode. The CDC, FDA and USDA Food Safety and Inspection Service (FSIS) with a full-time staff of FDA’s Coordinated Outbreak Response and Evaluation (CORE) network have continued their operations for monitoring and tracking any outbreak situation (FDA, 2020c).”
- One question though. Table 5 shows a list of recent recalls, but the newest one is 2 years old. We have not seen a recall since? If so, it would worthy of an explanation.
Ans. Thank you so much for pointing out the situation. When checked again a recent baby food recall occurred because of high arsenic levels in June 9, 2021 (Line 166). We have changed the table accordingly.
Reviewer 3 Report
This overview is timely given the rising importance of food safety and quality standards. The paper is well written and very easy to follow. My only comment is that it seems an important player in the field could be missing: private food safety standards including ISO 22000, BRC, SQF, PrimusGFS, GlobalGAP, IFS, FSSC22000. Over half of those standards are manufacturing standards and I was expecting to see some mentioning and discussion of those standards (also based on HACCP principle). If these standards do not apply, the authors want to be specific about this in this overview.
Author Response
We would like to thank you providing valuable feedback. Kindly see our reply to the comments.
This overview is timely given the rising importance of food safety and quality standards. The paper is well written and very easy to follow. My only comment is that it seems an important player in the field could be missing: private food safety standards including ISO 22000, BRC, SQF, PrimusGFS, GlobalGAP, IFS, FSSC22000. Over half of those standards are manufacturing standards and I was expecting to see some mentioning and discussion of those standards (also based on HACCP principle). If these standards do not apply, the authors want to be specific about this in this overview.
Response: We are thankful for your positive and encouraging feedback. We have added a table of said private food safety standards to make the manuscript more thorough.
Line 194-200
“Recently private standards have also emerged and evolved. Table 4 summarizes some of the private global food safety standards currently available for the industry. These private standards are now becoming progressively important.”
Line 205-207
Global Safety Standard Certification |
Description |
International Organization for Standardization (ISO 22000) |
The ISO 22000 standard describes the food safety management system requirements for any organization involved in the food chain such as ingredient producers, retailers, catering services, transportation etc. Any organization can pursue certification and registration if it conforms with this standard. |
Safe Quality Food (SQF) 1000/2000 |
It is a Hazard Analysis and Critical Control Points (HACCP) based certification system for food safety and quality of ingredients, packaging, farming, packing houses etc. |
British Retail Consortium (BRCGS) |
This standard has been developed in collaboration with the industry for provision of product safety and quality. |
PrimusGFS |
This certification program is farm focused Global Food Safety Initiative (GFSI). |
Global Good Agricultural Practices (GAP) |
This certification program covers agricultural crops primarily such as fruits, vegetables, hop, tea etc. |
International Featured Standards (IFS) |
This certification program covers the processes in the supply chain by doing risk-based assessment |
Food Safety Management Certification (FSSC) 22000 |
This certification is based on ISO 22000, ISO 9001, ISO/TS22003 and ISO 22003 and confirms food safety and quality is the organization is certified.
|